# Knowledge Management Process, Entrepreneurial Orientation, and Performance in SMEs: Evidence from an Emerging Economy

**Shiaw Tong Ha** [1], **May Chiun Lo** [1,*], **Mohamad Kadim Suaidi** [2], **Abang Azlan Mohamad** [1] **and Zaidi Bin Razak** [3]

1    Faculty of Economics and Business, Universiti Malaysia Sarawak, Kota Samarahan 94300, Malaysia; revetong@gmail.com (S.T.H.); maazlan@unimas.my (A.A.M.)
2    Chancellery, Universiti Malaysia Sarawak, Kota Samarahan 94300, Malaysia; kadim@unimas.my
3    Sarawak Multimedia Authority (SMA), Level 5, Kuching 93000, Malaysia; drzaidi@sma.gov.my
*     Correspondence: mclo@unimas.my

**Abstract:** Knowledge management (KM), a process of acquiring, converting, applying, and protecting knowledge assets, is crucial for value creation. The purpose of this research is to empirically test the relationship between KM processes (knowledge acquisition, knowledge conversion, knowledge application, and knowledge protection), entrepreneurial orientation (EO), and firm performance. Data were collected from 159 small- and medium-sized enterprises (SMEs) in Malaysia using a cross-sectional survey. This research uses partial least squares structural equation modeling (PLS-SEM) and WarpPLS version 7.0 to test the model. The results show that three of four KM dimensions: knowledge acquisition, knowledge conversion, and knowledge protection are positively related to performance. Moreover, EO has been found to moderate the relationship between knowledge application and performance positively.

**Keywords:** knowledge management (KM); knowledge acquisition; knowledge conversion; knowledge application; knowledge protection; entrepreneurial orientation (EO)

## 1. Introduction

In light of the coronavirus disease 2019 (COVID-19) outbreak, many businesses have been forced to close temporarily and shift to a remote work paradigm. As a result, cloud computing and online communications systems, such as Zoom, Microsoft Team, Google Meet, and Cisco Webex, are in higher demand around the world. These digital productivity tools have increased the volume of data generated from various sources, including business processes, social media platforms, sensor data, and machine-to-machine data. To remain competitive, a system that can capture, share, apply, and store these essential data is required. Hence, knowledge management (KM) appears to be more important than ever for organizational success.

KM is a discipline that involves the process of acquiring, converting, applying, and protecting a firm's information and knowledge assets [1]. By making data and information visible and accessible to organization members when needed, effective KM can help build new competitive advantages. While an increasing number of studies examine KM in relation to desired organizational outcomes, little is known about how different KM dimensions affect organizational outcomes. Instead, previous research has tended to consider KM as a composite construct by combining all of its dimensions into a single variable, thereby making it difficult to assess the impact of each dimension of KM on the organizational outcomes [2–4]. In this regard, Mills and Smith [5] contend that not all KM dimensions are directly related to organizational performance and warrant further study concerning this matter. Consistent with this perspective, Mohamad et al. [6] assessed the impacts of multiple dimensions of KM on firm innovativeness, and they found that

knowledge conversion was not positively related to innovativeness. As a result, a more comprehensive understanding of how each dimension of KM is linked to organizational outcomes is required.

Small- and medium-sized enterprises (SMEs) account for the majority of businesses globally, and they play an important role in GDP growth, job creation, and entrepreneurship. Despite their importance, existing literature shows that most KM research has been conducted in large organizations, while KM in SMEs is still at infancy stage and provides only fragmentary insight [7–9]. It is crucial to notice that SMEs are not a scaled-down replica of large organizations, and they are substantially different in many ways. SMEs, in comparison to large corporations, are typically less flexible in terms of human and financial resources. At the same time, SMEs have advantages over large organizations in that they are less bureaucratic, quick to change, and more flexible [10,11]. Hence, KM theories and practices that work well for large organizations may not be a good fit for SMEs [12,13]. Moreover, the limited resources of many SMEs prevent them from pursuing too many strategic options which would spread their resources too thinly. As a result, identifying which dimensions of KM can best boost organizational performance goals is critical in the short term and can be extended to improve on long term production and performance.

Extant literature has studied the performance contribution of KM; however, the results are mixed. Few scholars have reported significant and positive relationships between KM dimensions and desired organizational outcomes [14,15], while others have found an insignificant or indirect relationship between some KM dimensions and desired organizational outcomes [6,9,10]. These mixed results left the KM-performance debate open, and scholars have stressed the necessity for more research on the moderators to scrutinize the inconclusive results. The characteristics of entrepreneurial orientation (EO), such as innovativeness, proactiveness, and risk-taking, may facilitate KM development and lead to better utilization of knowledge resources [16,17]. When KM processes and EO work together, it is assumed that they will improve organizational strategy and assure organizational success [18–20].

To summarize, the relationship between particular KM dimensions and firm performance is still a point of contention in the literature, and researchers have emphasized the importance of more study on moderators to scrutinize the inconsistent results. Additionally, there is still a lack of consensus regarding the performance contribution of KM in the context of SMEs. Given the enormous number of SMEs and their significant contribution to economic development, this study aims to extend and integrate these streams of research and fill the aforementioned gaps by presenting a model that combines KM processes, EO, and the performance of SMEs. The study's findings are crucial for policymakers because they demonstrate which dimensions of KM best boost organisational performance goals.

The objectives of this study are twofold. First, this study investigates various dimensions of KM process capability and their effects on performance in the context of manufacturing-related SMEs. Second, it assesses whether EO moderates the relationships between KM process capabilities and performance. The rest of this paper is organized in the following manner. The theoretical foundation for this study is presented in Section 2, and the research methodology is presented in Section 3. Sections 4 and 5 contain the empirical findings. The implications, limitations, and future research directions are discussed in Section 6 of the paper.

## 2. Literature Review

### 2.1. Theoretical Underpinning

Knowledge-based view (KBV) and dynamic capability view (DCV) are the underlying theories of this study. KBV is an extension of the resource-based view. It indicates that a firm can gain a long-term competitive advantage by using its heterogeneous, socially complex, and difficult-to-imitate knowledge-based resources [21–23] DCV, on the other hand, claims that a firm's competitiveness is determined by its capability to integrate, build, and reconfigure internal and external competencies [24].

Knowledge is the most crucial resource for firms to establish a long-term competitive advantage and differentiate themselves from their competitors, according to KBV [21,25,26]. Given the importance of knowledge, firms must implement KM to successfully manage their knowledge. For this study, we follow Gold et al. [1], who proposed four dimensions of KM process capabilities as significant determinants of firm performance: knowledge acquisition, knowledge conversion, knowledge application, and knowledge preservation. Having knowledge assets, however, does not guarantee a competitive advantage in today's volatile business environment [2,27]. According to DCV, firms must integrate and build competencies to maximize the potential of their resources [24]. In this study, EO is identified as a firm's dynamic capability [19]. If KM and EO are aligned, it is expected that they will attain complementarities that will lead to superior firm performance. Figure 1 depicts the relationships between KM process capabilities, EO, and firm performance, which are described in the next subsections.

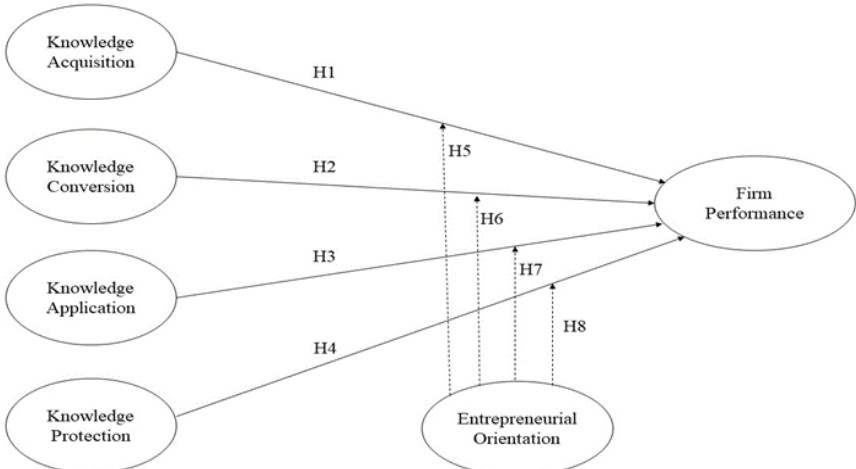

**Figure 1.** Research model.

### 2.2. Knowledge Acquisition and Firm Performance

If a firm wants to increase its understanding of consumer needs, the business environment, or rival actions, it must engage in knowledge acquisition, which is the process of gathering information from a variety of sources [1]. Firms must establish relationships with business partners, such as clients, suppliers, and group firms, in order to have access to the knowledge and information needed to produce productive and innovative operations [28,29]. By establishing a solid knowledge base, firms can improve their ability to respond effectively to changing market situations [8,30]. A solid knowledge base can aid businesses in making better decisions, lowering employee turnover, and maximizing market opportunities. Several lines of evidence suggest that a firm's competitive advantage and performance are heavily reliant on knowledge acquisition that improves the availability of the relevant knowledge to make the best judgments [31–33]. As a result, it is hypothesized that:

**Hypothesis 1 (H1).** *There is a positive relationship between knowledge acquisition and firm performance.*

### 2.3. Knowledge Conversion and Firm Performance

Knowledge conversion assists firms in making the greatest use of their knowledge by transforming individual knowledge into organizational knowledge [1]. The conversion of knowledge not only entails the act of converting tacit to explicit and explicit to tacit knowledge, but it also facilitates knowledge exchange [26]. This process is vital for the firms to avoid losing critical information or skills due to the departure of one or more employees. Therefore, firms must make reasonable efforts to build a culture that encourages employees

to create, store, and share their knowledge. Knowledge conversion has been defined in previous studies as a crucial variable that can contribute to creativity, innovation and ultimately, competitive advantage [34–36]. Knowledge conversion has also been shown to improve firm performance by creating important organizational knowledge and making it timely available in a shared database [15]. Hence, the following hypothesis is formulated:

**Hypothesis 2 (H2).** *There is a positive relationship between knowledge conversion and firm performance.*

### 2.4. Knowledge Application and Firm Performance

The next stage is to put the structured knowledge into practice. Knowledge application allows a firm to react more quickly to changing business conditions by incorporating knowledge into new products or processes [1,15]. Knowledge application, according to Alavi and Leidner [37], is the most significant KM process for improving organizational performance. Zaim et al. [38] backs up this claim, finding that knowledge application had the most substantial impact on improving KM-related organizational performance. On the other hand, knowledge application has been found to play a critical role in enhancing operational procedures and fostering better decisions, all of which contribute to improved business performance [32,33]. It is also a key determinant when it comes to innovation [35,39]. As claimed by Serrasqueiro et al. [40], R&D activities can make a considerable contribution to the growth of SMEs. Hence, the following hypothesis is developed:

**Hypothesis 3 (H3).** *There is a positive relationship between knowledge application and firm performance.*

### 2.5. Knowledge Protection and Firm Performance

Knowledge protection refers to a firm's ability to secure its intellectual knowledge from illegal theft and inappropriate use [1]. If a firm implements intellectual property protections, such as a patent, copyright, or trademark, knowledge protection will be more effective. These safeguards give the company the right to prevent competitors from copying its ideas or inventions, as well as the ability to benefit from licensing its intellectual property rights [39]. Knowledge protection is positively related to organizational performance in several studies. Ferri et al. [41], for example, offered empirical evidence that the patenting procedure has a positive impact on spin-off performance. Similarly, Liu and Deng [15] discovered that knowledge protection improves business process outsourcing performance since other companies are unable to quickly copy the ideas or inventions. Based on the preceding arguments, the following hypothesis is formulated:

**Hypothesis 4 (H4).** *There is a positive relationship between knowledge protection and firm performance.*

### 2.6. The Moderating Effects of EO

Since its inception in 1970, EO has become one of the most widely studied entrepreneurship and management concepts. EO refers to the decision-making styles, management practices, and behaviors characterized by innovative, proactiveness, and risk-taking [42–44]. Creating an EO culture will assist firms in identifying and exploiting new possibilities, creating new values, and becoming market leaders. Not only that, but EO is seen as a critical driver of a company's competitiveness and performance, especially in dynamic business environments [45–47].

Furthermore, EO has been found to interact with other organizational factors, such as manufacturing capabilities, to improve organizational performance [48]. A firm may seek to develop flexibility and cost leadership strategies, such as investment in technology and automated processes, to strengthen its competitive position by having greater EO. The moderating effect of EO was also discovered by Yousaf and Majid [49], who discovered that EO strengthens the relationship between organizational flexibility and strategic business performance. The rapid development of new products and services, propensity to intensely

challenge competitors, and greater risk-taking behaviors resulting from a greater degree of EO can boost firms' flexibility and improve the strategic business performance.

On the other hand, EO directs firms towards resource leveraging to unlock new market opportunities [19,50]. Wiklund and Shepherd [17] discovered that EO moderates the positive relationship between knowledge-based resources and SMEs' performance. The authors argue that, when EO is low, firms may be reluctant to maximize the utilization of knowledge-based resources by perceiving such resources as less important. Under the EO culture, firms are motivated to configure knowledge-based resources into commercially valuable resource bundles. These valuable resource bundlers can help the firms achieve a higher absorptive capacity level that results in superior performance [51].

EO is also expected to strengthen KM [52,53]. Knowledge acquisition, conversion, application, and protection are all critical KM processes for improving business performance. However, equally important are the positive mindset attributes to apply such processes to reach the organizational goals. It is known that EO requires firms to be more innovative, risk-taking, and proactive in their operations to identify and exploit new market opportunities. These positive mindset attributes may motivate the firms to develop greater KM processes in order to create more innovative products or services that the competitors will not be able to match or exceed. When EO is low, on the other hand, a firm may be hesitant to develop KM processes at a higher level which are often risky and costly. In light of the foregoing discussion, the following hypotheses regard the moderating effect of the EO in the KM-firm performance relationship are postulated:

**Hypothesis 5 (H5).** *EO positively moderates the relationship between knowledge acquisition and firm performance.*

**Hypothesis 6 (H6).** *EO positively moderates the relationship between knowledge conversion and firm performance.*

**Hypothesis 7 (H7).** *EO positively moderates the relationship between knowledge application and firm performance.*

**Hypothesis 8 (H8).** *EO positively moderates the relationship between knowledge protection and firm performance.*

## 3. Research Method

### 3.1. Data Collection and Sample Characteristics

Primary data and quantitative approaches were employed to test the proposed model. A survey was conducted among a randomly selected sample of manufacturing-related SMEs with full-time employees ranging from 5 to 200 and annual sales revenue ranging from RM300,000 to RM50 million. The three states of Malaysia, namely Selangor, Johor, and Sarawak, were selected for this study because they are the top three recipients in terms of the number of approved manufacturing projects at the time of the study, according to data from the Malaysian Investment Development Authority (MIDA). The population for this study is 1011 manufacturing SMEs that are listed in the Federation of Malaysian Manufacturers (FMM) Directory 2015. As shown in Table 1, 114 SMEs were selected as the sample by using a proportionate stratified sampling method.

**Table 1.** The distribution of SMEs by locations.

| Strata | No of Population | Proportionate Ratio | Sample Size of Each Strata |
|---|---|---|---|
| Selangor | 766 | 114 (766/1011) | 86 |
| Johor | 215 | 114 (215/1011) | 24 |
| Sarawak | 30 | 114 (30/1011) | 4 |
| **Total** | 1011 | 114 (1011/1011) | 114 |

Data were collected in two stages. First, a pretesting with ten SMEs was undertaken in March 2019 to ensure that the respondents understood the questions and that there were no issues, such as unclear language or an incorrect order, that could lead to biased responses. The ten participants in the pretesting and their responses were not included in the final sample. Some survey items were carefully revised and polished based on comments from the pretesting. A total of 600 questionnaires was distributed to the leaders of SMEs with positions, such as Chief Executive Officer, President, Chairman, Managing Director, or General Manager, along with a cover letter explaining the study's objectives. The data collection process took two months (April and May 2019), with 171 respondents returning responses. A sample of 159 respondents was deemed fit for final analysis after data screening, yielding a response rate of 26.5%. To verify this sample size, a G*power analysis was conducted The minimum sample size generated by the G*power is 114 ($f^2 = 0.15$, $\alpha$ criterion = 0.05, power = 0.8). With 159 responses, the data analysis appears to have sufficient power.

Sample characteristics show that most of the respondents were male (71.1%). The majority of the respondents hold a bachelor's degree (47.2%) and diploma (33.3%), while 20% have completed high school level education, and 6.9% hold a postgraduate degree. Most of the respondents were aged between 22–30 years old (40.9%), 37.7% were aged between 31–40 years old, 14.5% were aged between 41–50 years old, and only 6.9% were aged 51 years old and older. This figure may imply a tendency that the employees participate less and less when they get older. KM is crucial in this case, as it involves the processes that facilitate the knowledge transfer between employees.

### 3.2. Measures

The constructs and associated measures used in this study are listed in the Appendix A. All of the constructs were operationalized using reflective measures, which were adapted from previous studies with minor modifications to make them more relevant to the SMEs context. Using the scales developed by Gold et al. [1], 16 items were adapted to assess knowledge management processes across four dimensions: acquisition (four items), conversion (four items), application (four items), and protection (four items). The five-item scale developed by Liu et al. [19] was used to operationalize EO, which incorporates all significant EO components into a single concept that determines an organization's efficiency in organizing and reconfiguring its resources. Five items adapted from Prieto and Revilla's study [54] were used to assess firm performance. Because objective performance data is usually unavailable or difficult to get in the case of SMEs, subjective performance measures are considered to be a more appropriate approach [25,55]. All items were measured on a 7-point Likert scale, with 1 indicating "strongly disagree" and 7 indicating "strongly agree."

### 3.3. Data Screening

Before moving on to statistical analysis, the data were screened to ensure their integrity. In this study, the data screening involves three procedures: missing value analysis, identifying suspicious response patterns, and testing for common method bias.

As an initial step in data screening, responses were screened for missing data. Following Hair et al. [56], an observation should be removed from the data file if it has missing data of more than 15%. Of the 171 responses received, 11 of them were removed due to the amount of missing data of more than 15%. In addition, to effectively remedy missing values, Little's MCAR (missing completely at random) test was applied using SPSS software. The results show that missing data are random ($\chi2 = 365.601$, df = 326, $p = 0.064$); therefore, expectation-maximization (EM) algorithm [57] was used to impute missing values in this study.

The data was then screened for suspicious response patterns, such as inconsistencies in answers and straight-lining. No cases were identified to have inconsistencies in answers. However, in one case, the responses were found to be straight-lining. Straight-lining is supported by the fact that the particular respondent gave the same answer (neutral) for all

questions. Most researchers agree that a straight-lining response may pose a severe threat to the data quality; therefore, this case was eliminated [56,58].

Because this study used a single informant to collect data, bias due to common method variance might undermine the constructs' actual relationship. To minimize the threat of common method bias, both procedural and statistical remedies were adopted [59]. Procedural remedies were applied before data collection, such as protecting respondent's anonymity, assuring the respondents that there were no right or wrong answers, and only aggregate data was used. Statistical remedies were taken after data collection to evaluate the extent of common method bias by conducting two different tests. First, Harman's single factor test was conducted by including all items into a maximum likelihood factor analysis [60]. With all items included, three factors were extracted, explaining a total of 61.55% variance. No single factor was found to account for more than 50% of the covariance, showing no evidence of common method bias. Next, the full collinearity variance inflation factor (VIF) was calculated using WarpPLS 7.0 software [61]. Common method bias is said to be present if the full collinearity VIF value is more than 5 [62]. The full collinearity VIF values for knowledge acquisition (3.423), knowledge conversion (2.907), knowledge application (2.901), knowledge protection (2.640), EO (1.847), and performance (3.519) ascertained no substantial common method bias. Taken together, these tests confirmed that common method bias was not a significant threat in this study. After data screening, a final sample of 159 responses was used for statistical analysis.

## 4. Results

### 4.1. Assessment of the Measurement Model

Construct's reliability and validity were examined by assessing the measurement model using WarpPLS 7.0 software. Table 2 presents the criteria used to evaluate reliability and validity. The factor loadings, average variance extracted (AVE), and composite reliability (CR) were used to test the convergent validity. The value of factor loadings should exceed 0.70, AVE should exceed 0.50, and CR should exceed 0.70 by following the recommendations of Hair et al. [63]. As shown in Table 2, all factor loadings, AVE, and CR were within the acceptable range, suggesting good convergent validity. Next, constructs' reliability was examined by using CR and Cronbach's alpha. The CR and Cronbach's alpha values for all constructs were found to exceed 0.70 and below 0.95, thereby denoting the reliability of the measurement model.

**Table 2.** Summary of construct reliability and validity.

| Construct | Items | Loadings | AVE | CR | Cronbach's Alpha |
|---|---|---|---|---|---|
| Knowledge Acquisition | KAQ1 | 0.795 | 0.712 | 0.908 | 0.865 |
| | KAQ2 | 0.873 | | | |
| | KAQ3 | 0.851 | | | |
| | KAQ4 | 0.854 | | | |
| Knowledge Conversion | KC1 | 0.755 | 0.664 | 0.887 | 0.830 |
| | KC2 | 0.876 | | | |
| | KC3 | 0.798 | | | |
| | KC4 | 0.826 | | | |
| Knowledge Application | KAP1 | 0.843 | 0.755 | 0.925 | 0.891 |
| | KAP2 | 0.896 | | | |
| | KAP3 | 0.891 | | | |
| | KAP4 | 0.843 | | | |
| Knowledge Protection | KP1 | 0.844 | 0.722 | 0.912 | 0.872 |
| | KP2 | 0.874 | | | |
| | KP3 | 0.872 | | | |
| | KP4 | 0.808 | | | |
| Entrepreneurial Orientation | EO1 | 0.829 | 0.750 | 0.938 | 0.917 |
| | EO2 | 0.901 | | | |
| | EO3 | 0.888 | | | |
| | EO4 | 0.854 | | | |
| | EO5 | 0.858 | | | |
| Firm Performance | FP1 | 0.742 | 0.610 | 0.887 | 0.840 |
| | FP2 | 0.795 | | | |
| | FP3FP4 | 0.8200.801 | | | |
| | FP5 | 0.746 | | | |

To assess the constructs' discriminant validity, Fornell and Larcker [64]'s criterion was used. As presented in Table 3, the AVE's square root for each construct (represented by the bold values) was found to be greater than the absolute value of inter-construct correlations, showing no evidence of a lack of discriminant validity. Besides constructs reliability and validity, multicollinearity was also assessed by calculating average block variance inflation factor (AVIF) and average full collinearity VIF (AFVIF) [61,62]. Following the recommendation from Kock [62], both AVIF and AFVIF values should be equal to or lower than 3.3 to negate the existence of multicollinearity. The results (AVIF = 2.737; AFVIF = 3.104) ascertained the non-existence of multicollinearity.

**Table 3.** Discriminant validity of constructs.

|  | 1 | 2 | 3 | 4 | 5 | 6 |
|---|---|---|---|---|---|---|
| 1. Knowledge Acquisition | **0.844** |  |  |  |  |  |
| 2. Knowledge Conversion | 0.764 | **0.815** |  |  |  |  |
| 3. Knowledge Application | 0.735 | 0.685 | **0.869** |  |  |  |
| 4. Knowledge Protection | 0.709 | 0.696 | 0.691 | **0.850** |  |  |
| 5. Entrepreneurial Orientation | 0.459 | 0.442 | 0.488 | 0.417 | **0.866** |  |
| 6. Firm Performance | 0.601 | 0.586 | 0.586 | 0.596 | 0.604 | **0.781** |

### 4.2. Assessment of the Structural Model

This study performed partial least squares structural equation modeling (PLS-SEM) to test the hypotheses. Table 4 and Figure 2 present the results of the hypotheses testing. The results show that knowledge acquisition ($\beta$ = 0.197, $p < 0.05$) had a significant positive relationship with firm performance (H1). Similarly, knowledge conversion ($\beta$ = 0.266, $p < 0.001$) and knowledge protection ($\beta$ = 0.162, $p < 0.05$) were found to be positively related to firm performance (H2 and H4, respectively). In addition, EO ($\beta$ = 0.179, $p < 0.05$) was found to moderate the relationship between knowledge application and firm performance (H7). In terms of effect sizes ($f^2$), all constructs had shown weak (0.017) to medium (0.164) effects on firm performance, except the moderating effect of EO on knowledge protection-firm performance relationship ($f^2$ = 0.000).

**Table 4.** Summary of path coefficients and hypotheses testing.

| Hypotheses | Relationship | $\beta$ | *p*-Value | $f^2$ | Decision |
|---|---|---|---|---|---|
| | | Direct Relationships | | | |
| H1 | KAQ -> FP | 0.197 | 0.005 * | 0.122 | Supported |
| H2 | KC -> FP | 0.266 | <0.001 ** | 0.164 | Supported |
| H3 | KAP -> FP | 0.125 | 0.053 | 0.076 | Not Supported |
| H4 | KP -> FP | 0.162 | 0.018 * | 0.097 | Supported |
| | | Moderating Effects of Entrepreneurial Orientation | | | |
| H5 | EO*KAQ -> FP | 0.118 | 0.064 | 0.020 | Not Supported |
| H6 | EO*KC -> FP | 0.078 | 0.160 | 0.017 | Not Supported |
| H7 | EO*KAP -> FP | 0.179 | 0.010 * | 0.048 | Supported |
| H8 | EO*KP -> FP | −0.002 | 0.491 | 0.000 | Not Supported |

Note: * $p < 0.05$, ** $p < 0.001$.

The predictive accuracy of the model was examined by calculating the coefficient of determination ($R^2$). As shown in Figure 2, the four exogenous constructs: knowledge acquisition, knowledge conversion, knowledge application, and knowledge protection combined explained 54% of the variation in firm performance. Referring to Hair et al. [63], the $R^2$ value for firm performance (0.54) was considered moderate. To assess predictive relevance, Stone-Geisser's $Q^2$ value was used [65,66]. Following the suggestion by Hair et al. [56], a blindfolding procedure was used to assess the impact of KM on firm performance. The results show that the $Q^2$ value of firm performance was 0.515, which is indicative of the large effect of the four KM processes [63,67,68].

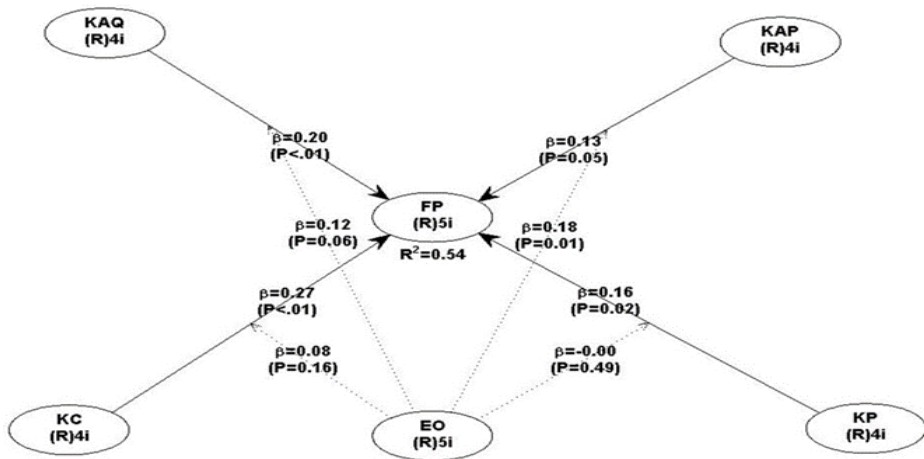

**Figure 2.** Results of the path analysis.

## 5. Discussion

The purpose of this study was to provide empirical evidence on how a specific KM process supports performance and whether EO moderates the positive relationship between KM processes and performance in SMEs. By decomposing KM into four processes, this study addresses Mills and Smith [5]'s call for KM to be considered as a multidimensional construct to effectively evaluate KM's consequences. In addition, this study views EO as an important complementary asset that can strengthen the relationship between KM processes and firm performance by positively leveraging its moderating effect. The major findings of this study are interpreted and discussed below.

Consistent with prior studies, the results indicate that knowledge acquisition has a positive and significant relationship with firm performance [32,33]. To withstand today's dynamic business environment, firms must place great emphasis on knowledge acquisition. This process enables firms to continuously expand and update their existing knowledge base by identifying and acquiring valuable know-how; thus, firms can improve their responsiveness to market change and performance. Therefore, firms should establish relationships with business partners, such as clients, suppliers, and group firms, in order to have access to the knowledge and information needed to generate new knowledge and develop innovative advances [28,29].

Furthermore, the results revealed that knowledge conversion is significantly and positively related to firm performance, and this relationship is the strongest in the model. This finding corroborates previous studies that view knowledge conversion as a core capability to enhance firm performance [15,34]. Firms need to develop a framework for organizing, integrating, or disseminating knowledge as these processes enable firms to reduce redundancy and replace obsolete knowledge, which represents a key to achieving superior performance.

As predicted, knowledge protection is found to be positively and significantly related to firm performance. This finding is consistent with prior research stressing the benefits of building mechanisms for securing knowledge assets of the firms [41]. It appears that firms with strong knowledge protection capability can protect their proprietary knowledge from being illegally or inappropriately used by others (inside and outside firms). Thus, these firms can sustain their performance for a longer period.

One interesting finding of this study is that, among the four process capabilities of KM, only knowledge application was not significantly related to firm performance. This finding is supported by some previous studies, which found that not all KM capabilities are directly related to firm performance [5,6]. The insignificant effect of knowledge application on performance might be explained by less financial and administrative resources in SMEs [69]. Most SMEs tend to employ one or two employees to hold the firms' key knowledge due

to resource constraints. Consequently, it may impede the flow of knowledge and hamper innovative actions, resulting in the loss of a valuable commercial opportunity.

Furthermore, EO has demonstrated its ability to act as a moderator in the relationship between KM processes and SMEs performance. Through EO, the relationship between knowledge application and performance becomes stronger. This finding backs up Wiklund and Shepherd [17]'s findings that EO can improve firm performance by maximizing the utilization of knowledge-based resources. It appears that EO, which is defined by risk-taking, inventiveness, and proactiveness, may inspire businesses to share and utilize knowledge in order to exploit new opportunities [51,53]. Taking EO as an organizational climate, this finding backs up Li et al. [52]'s claim that EO can aid the firms in achieving superior performance by enhancing knowledge application quality.

## 6. Conclusions

This study has provided empirical evidence on the differential impacts of KM dimensions on SMEs performance. Knowledge acquisition, conversion, and protection were all positively related to the performance of SMEs. Surprisingly, no significant and positive relationship was found between knowledge application and the performance of SMEs. In addition, the moderating effects of EO have been discovered. The relationship between knowledge application and the SMEs performance was found to be moderated by EO. This study has provided a beneficial guideline for practitioners on what is the best KM capability that can be used to improve firm performance and how EO may be paired with KM application to produce superior performance.

### 6.1. Theoretical and Practical Implications

This study contributes to the body of knowledge on KM and EO and provides important implications for practitioners involved in the firms' migration toward KM. Composite measures are frequently used when examining the KM-performance link, making it difficult to understand how particular KM dimensions relate to firm performance [2,3]. This study fills the gap by developing and empirically testing an integrated research model that assesses differential impacts of KM process capabilities on firm performance. Findings show that three of the four KM process capabilities: knowledge acquisition, conversion, and protection, have a significant positive relationship with firm performance. These findings corroborate with Mills and Smith [5], in that KM dimensions have varying effects on performance and that not all KM dimensions are directly related to performance.

Another notable contribution of this study is that it examined EO as a moderator that was otherwise considered an independent variable traditionally and answered whether EO positively moderates the relationships between KM processes and SMEs performance. In this sense, the study's findings demonstrated that EO characteristics, such as risk-taking, innovativeness, and proactiveness, are critical to the effective deployment of knowledge application. These findings shed light on how a firm might combine a dynamic capability (in this case, KM) with another (in this case, EO) to obtain greater results.

The main practical implication of this study is that it provides some insights for SMEs on how to improve their performance through KM processes. Such insights can assist policymakers in making better investment decisions and increasing the success of their KM programme. Results of this study inform the policymakers of SMEs that knowledge acquisition, conversion, and protection contribute to firm profitability. Among the three KM processes, knowledge conversion that creates the right knowledge and makes knowledge easily accessible is of utmost importance. Therefore, knowledge conversion should be prioritized within the SMEs, and a culture of creating and sharing knowledge must be built into the DNA of the SMEs.

Second, the findings of this study help policymakers understand the critical role of EO in strengthening the positive effect of knowledge application on firm performance. Likely, old thinking patterns and fear of trying new things will inherently fail to create innovations. Hence, policymakers themselves and employees must be more innovative,

proactive, and prone to taking calculated risks in order to generate creative ideas that may drive firm growth. The Malaysian government offers a comprehensive range of programs, such as the SME Technology Transformation Fund (STTF), to assist SMEs with access to digital and technology options. SMEs should use these grants to implement digitalization in their operations in order to enhance their innovation capability.

### 6.2. Limitations and Future Research Suggestions

Despite this study's contributions, several limitations require attention and need further assessments. First, the data used in the analyses were collected from a single source. Although several procedural and statistical remedies have been adopted to minimize the possibility of bias, future research could consider collecting data from multiple respondents within a firm. Second, future studies could look into other aspects of KM, such as technology, structure, and culture. Third, this study is limited to a single context: Malaysian manufacturing SMEs. Other sectors, such as the service sector, tourism sector, and agriculture sector, where KM implementation is still understudied, need to be investigated in conjunction with currently studied variables. At the same time, future research could consider conducting a comparative study to test whether this model has the same implications for SMEs in other countries. Future research should consider these suggestions to cross-validate and conclude the interrelationships between KM, EO, and firm performance. Finally, the questionnaire items relating to the impact of COVID-19 on business process management are not included in this study. The contribution to the existing literature could have been more outstanding if this study had considered the impact of COVID-19 in businesses. Researchers are encouraged to conduct more research to identify successful KM practices that can establish and maintain performance and competitive advantage during the pandemic.

**Author Contributions:** Introduction, S.T.H. and M.C.L.; research framework, S.T.H., M.C.L. and A.A.M.; data collection, S.T.H.; data analysis, S.T.H., M.C.L. and A.A.M.; resources, M.K.S., Z.B.R.; writing—original draft preparation, S.T.H. and M.C.L.; writing—review and editing, M.K.S., Z.B.R. and A.A.M.; supervision, M.C.L., M.K.S., Z.B.R. and A.A.M.; funding acquisition, M.C.L., M.K.S. and Z.B.R. All authors have read and agreed to the published version of the manuscript.

**Funding:** This research was funded by Universiti Malaysia Sarawak.

**Institutional Review Board Statement:** Not applicable.

**Informed Consent Statement:** Not applicable.

**Data Availability Statement:** Data sharing not applicable.

**Conflicts of Interest:** The authors declare that they have no conflict of interests.

### Appendix A

**Table A1.** Questionnaire items.

| Construct/Item | |
|---|---|
| Knowledge Acquisition | |
| KAQ1 | My organization has processes for acquiring knowledge about our customers. |
| KAQ2 | My organization has processes for exchanging knowledge with our business partners. |
| KAQ3 | My organization has processes for acquiring knowledge about competitors within our industry. |
| KAQ4 | My organization has processes acquiring knowledge about new products or services within our industry. |

**Table A1.** *Cont.*

| | Construct/Item |
|---|---|
| | **Knowledge Conversion** |
| KC1 | My organization has processes for converting knowledge into design of new products or services. |
| KC2 | My organization has processes for integrating different sources and types of knowledge. |
| KC3 | My organization has processes for distributing knowledge throughout the organization. |
| KC4 | My organization has processes for converting knowledge competitive intelligence into plans of action. |
| | **Knowledge Application** |
| KAP1 | My organization has processes for applying knowledge learned from experience. |
| KAP2 | My organization has processes for using knowledge in development of new products or services. |
| KAP3 | My organization uses knowledge to adjust strategic direction. |
| KAP4 | My organization takes advantage of new knowledge. |
| | **Knowledge Protection** |
| KP1 | My organization has processes to protect knowledge from inappropriate use inside the organization. |
| KP2 | My organization has processes to protect knowledge from theft from within the organization. |
| KP3 | My organization has processes to protect knowledge from theft from outside the organization. |
| KP4 | My organization clearly identifies and communicates the importance of protecting knowledge which is restricted. |
| | **Entrepreneurial Orientation** |
| EO1 | My organization has a strong emphasis on R&D and technological leadership. |
| EO2 | My organization is willing to adopt a very competitive posture toward our competitors. |
| EO3 | My organization is willing to initiate a competitive response to the action taken by our competitors. |
| EO4 | My organization is aggressive when facing opportunities. |
| EO5 | My organization has a strong tendency to pursue high-risk projects. |
| | **Firm Performance** |
| FP1 | My organization has enhanced return on assets for the past few years. |
| FP2 | My organization has sales growth for the past few years. |
| FP3 | My organization has been profitable for the past few years. |
| FP4 | My organization has improved in work productivity for the past few years. |
| FP5 | My organization has improved in production cost for the past few years. |

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
