# Peer review of "Knowledge Management Process, Entrepreneurial Orientation, and Performance in SMEs: Evidence from an Emerging Economy"

_sustainability, doi:10.3390/su13179791_

Round 1

Reviewer 1 Report

Dear Editor,

Thank you for offering me the opportunity to review this study.
The work is well done, clear and orderly. The figures and tables are clearly described, the conclusions are consistent with the literature.
It seems that the authors are expert researchers, and this facilitates the reading of the paper. 

Nevertheless, the paper's contribution to the existing literature could have been more outstanding if it had considered the impact of COVID-19 in the specific research area under study. For this reason, I believe that the overall value of the paper is not high, even if, I repeat, well structured.

I also found some inconsistencies that require clarification. I report them below.

In line 212, the authors write: "(...) was used to select 600 manufacturing SMEs as a sample (...)". and again, in row 225 "Out of 600 SMEs in the sample" but in table 1, it appears that the initial sample is made up of 1,011 companies. Verify and clarify this aspect.

The authors do not specify the time in which the firms in the sample were observed. This data is essential to validate the results. Furthermore, in lines 31-34, the authors write: "To remain competitive, a system that can capture, share, apply, and store these essential data is required. In this case, knowledge management (KM) appears to be more important than ever for organizational success" referring to the impact of COVID-19 on business process management. No question in the questionnaire relates to this aspect, and this is a limitation of the research. I suggest the authors indicate it as such and foresee it as a future step of the research.

Good luck to the authors for their work!

Reviewer 2 Report

General comment:

The manuscript addresses a relevant topic, that is, the set of relationships between the knowledge management process and performance in SMEs. Moreover, the moderator role played by the entrepreneurial orientation of SMEs in this same relationship is analyzed. The theoretical argumentation is well-grounded in previous literature, albeit it needs to be reinforced with related studies. The empirical approach is considered robust and straightforward to follow. Nevertheless, the discussion of the new evidence needs to be reinforced by contrasting the previous related empirical studies. The concluding remarks need to be expanded, by providing both theoretical and managerial implications, as well as guidelines for future research.

Specific comments:

In order to increase the global quality of the manuscript several recommendations are made available below:

  1. Bearing in mind the relevant literature on growth and sustainable performance of SMEs, in the introductory item, the caveat found in the literature and innovative contributions should be outlined.
  2. The following two reference studies need to be incorporated in the literature review and in discussing the empirical findings:
    1. DOI: 10.1504/IJESB.2009.023357
    2. ISSN: 14502887
  1. Section 5. Discussion needs to be revised and reinforced using the 4 reference studies and other relevant studies published in your target journal.
  2. The concluding remarks need also to incorporate both theoretical and managerial implications, as well as guidelines for future research.

Round 2

Reviewer 2 Report

Congratulations on the revised version of the manuscript.